# Exploring the Bioactive Potentials of C_60_-AgNPs Nano-Composites against Malignancies and Microbial Infections

**DOI:** 10.3390/ijms23020714

**Published:** 2022-01-10

**Authors:** Kunal Biswas, Awdhesh Kumar Mishra, Pradipta Ranjan Rauta, Abdullah G. Al-Sehemi, Mehboobali Pannipara, Avik Sett, Amra Bratovcic, Satya Kumar Avula, Tapan Kumar Mohanta, Muthupandian Saravanan, Yugal Kishore Mohanta

**Affiliations:** 1Centre for Nanoscience and Nanotechnology, Sathyabama Institute of Science and Technology (Formerly Sathyabama University), Jeppiar Nagar, Salai, Chennai 600119, India; kunal.sapiens@gmail.com; 2Department of Biotechnology, Yeungnam University, Gyeongsan 38541, Gyeongsangbuk-do, Korea; awdhesh@ynu.ac.kr; 3Department of Biological Sciences, Asian Institute of Public Health University, Bhubaneswar 751002, India; pradiptaranjan.s08@gmail.com; 4Department of Chemistry, King Khalid University, Abha 61413, Saudi Arabia; agsehemi@kku.edu.sa (A.G.A.-S.); mpannipara@kku.edu.sa (M.P.); 5Department of Electronics and Electrical Communication Engineering, Indian Institute of Technology Kharagpur, Kharagpur 721302, India; aviksett210290@gmail.com; 6Department of Physical Chemistry and Electrochemistry, Faculty of Technology, University of Tuzla, Univerzitetska 8, 75000 Tuzla, Bosnia and Herzegovina; amra.bratovcic@untz.ba; 7Natural and Medical Sciences Research Centre, University of Nizwa, Nizwa 616, Oman; chemisatya@unizwa.edu.om (S.K.A.); tapan.mohanta@unizwa.edu.om (T.K.M.); 8Department of Pharmacology, Saveetha Dental College, Saveetha Institute of Medical and Technical Sciences, Chennai 600077, India; 9Department of Applied Biology, University of Science and Technology Meghalaya, Ri-Bhoi 793101, India

**Keywords:** fullerene, AgNPs, breast adenocarcinoma, antimicrobials, antioxidants, in-vitro

## Abstract

At present, the potential role of the AgNPs/endo-fullerene molecule metal nano-composite has been evaluated over the biosystems in-vitro. The intra-atomic configuration of the fullerene molecule (C_60_) has been studied in-vitro for the anti-proliferative activity of human breast adenocarcinoma (MDA-MB-231) cell lines and antimicrobial activity against a few human pathogens that have been augmented with the pristine surface plasmonic electrons and antibiotic activity of AgNPs. Furthermore, FTIR revealed the basic vibrational signatures at ~3300 cm^−1^, 1023 cm^−1^, 1400 cm^−1^ for O-H, C-O, and C-H groups, respectively, for the carbon and oxygen atoms of the C_60_ molecule. NMR studies exhibited the different footprints and magnetic moments at ~7.285 ppm, explaining the unique underlying electrochemical attributes of the fullerene molecule. Such unique electronic and physico-chemical properties of the caged carbon structure raise hope for applications into the drug delivery domain. The in-vitro dose-dependent application of C_60_ elicits a toxic response against both the breast adenocarcinoma cell lines and pathogenic microbes. That enables the use of AgNPs decorated C_60_ endo fullerene molecules to design an effective anti-cancerous drug delivery and antimicrobial agent in the future, bringing a revolutionary change in the perspective of a treatment regime.

## 1. Introduction

Fullerenes (C_60_) are caged zero-dimensional carbon nanostructures with their electrons confined in their native structure. The electrons in their structures are restricted to moving in all three directions along the axes of x, y, and z, respectively. The degree of freedom of electrons dictates the movement of underlying electrons in the structures that govern the myriad differential applications from electronics to the bio-medical field [1]. On the contrary, the silver nanoparticles (AgNPs) are metallic nanostructures with a huge surface-to-volume ratio and possess the number of surface electrons and overall surface energy at their surfaces [2,3]. It has been extensively reported that owing to the huge amount of surface electrons and surface energy on the nanoparticle surface, the effective interface of the silver nanoparticles with their biological counterparts increases exponentially [4]. Such silver nanoparticles and in composite forms with the interacting biological systems make the bio-nanocomposites a unique platform for several biological applications to disease diagnosis. The available surface electrons over the surface of AgNPs make a strong redox interaction with the available C_60_ electrons in its structure, making the interacting AgNPs with carbon-based structure a novel composite system at the nanoscale. In this study, carbon nanostructures were selected as the Buckminsterfullerene of polyhedral structural configuration, which has sixty-eighty carbon atoms in its native structure. The C_60_ molecule is a typical football-shaped structure with carbon (C) atoms in the pentagon and hexagon configurations. It has been named after the well-known scientist Buckminster Fuller, resembling its structure to geodesic domes [5,6,7,8,9,10,11]. It has been previously reported that C_60_ molecules can be prepared using bottom-up techniques using arch discharge, chemical route of Chemical Vapour Deposition (CVD), etc., where refined structures of C_60_ when visualized under High electron microscopy such as HR-TEM, FE-SEM could be employed for visualizing and analyzing its native configurations. When tested in biological systems, the crystalline structure of the molecule is also a paramount factor responsible for its differential toxicity responses. It has been speculated previously that when C_60_ molecules are tested in common cell lines, they exhibit differential toxicity responses when introduced at different dosages in in-vitro conditions [12,13,14,15,16,17].

In this study, the intrinsic metal toxicity of AgNPs when decorated over the C_60_ molecule has been tested in the biological systems of the breast adenocarcinoma (MDA-MB-231) cell line and normal human epidermal (HaCaT) cell line in an in-vitro situation. The C_60_-AgNPs nano-composites have been evaluated for their antimicrobial and antioxidant activities, apart from the anticancer activity test. The cell lines have been used as a biological target in laboratory conditions, with various dosages of the pristine C_60_, pristine AgNPs, and the combinations of C_60_-AgNPs nano-composites. These have been embarked for analyzing the combinatorial effect of both the molecular entities over the biological systems to determine a safer dosage in the effective dosimetry and may be helpful in the development of therapeutic nanoscaled material for the treatment of fatal diseases.

## 2. Materials and Methods

### 2.1. Materials

The chemicals used in the current experiments were: Fullerenes (C_60_) and AgNPs were procured commercially from Sigma-Aldrich (St. Louis, MO, USA) of 99.99 % purity grade with no further purifications. The solvent used in this study was double-distilled water. The other chemicals, such as Muller Hinton Agar (MHA) and Potato Dextrose Agar (PDA) media, were purchased from Hi-Media. Moreover, DPPH, streptomycin sulfate, benzyl antibiotics, and doxorubicin were purchased from Sigma Aldrich. The cell line media such as Dulbecco’s Modified Eagle’s Medium (DMEM), Trypsin-EDTA and Fetal Bovine Serum (FBS) were purchased from GIBCOTM (Grand Island, NY, USA).

### 2.2. Methods

The C_60_ molecule and the AgNPs were co-mixed using a conventional chemical route in a reagent vessel in laboratory conditions using a standard mixing protocol with slight modifications [18]. The mixed as formed nano-composites in the liquid forms are being stored in a hot-air oven for 12 h for drying. The prepared dried powders of the nano-composites were then being subjected to various characterizations using different physical and biophysical techniques like FT-IR, NMR, FE-SEM, and HR-TEM analysis. 

#### 2.2.1. FTIR Analysis

The as-prepared nano-composites of C_60_-AgNPs upon subjection to Fourier Transform Infra-Red Spectroscopy (FT-IR) were measured in the scanning range of 400–4000 cm^−1^. The samples were prepared using the Potassium Bromide (KBr) pellet and measured in the FT-IR instrumentation (Perkin–Elmer). The nature of the vibrational spectroscopic groups and the available functional groups were studied, prevalent in the nano-composite structure using this sophisticated technique.

#### 2.2.2. NMR (Nuclear Magnetic Resonance) Analysis

The as-prepared nano-composites were subjected to solid-state NMR spectrometer ^13^C-NMR (Jeol, 400 MHz), whereby the available magnetic moment in terms of ppm was analyzed in the native structure of the nano-composites. The net available AgNPs magnetic moment and the C60 partial moment were subjected for NMR analyses for uncovering the composite behavior and nature at the nanoscale for its different magnetic attributes and alignment.

In order to prepare the as-prepared nano-composite powder of C_60_-AgNPs for ^13^C NMR instrumentations characterizations, the solid powder of C_60_-AgNPs was packed manually in the 3.2 mm zirconia rotor. In brief, for the ^13^C measurements, the composite powder of C_60_-AgNPs was subjected to the one-pulse acquisition at MAS of 10 kHz. Technically, ^13^C π/2 pulse length has been employed ~5 µs. An approximate recycle delay for ~9 sec has been employed with an average acquisition time of ~16 ms for 5120 transients.

Additionally, during the characterization, a linear ramp of 100% on ^1^H channel for ^13^C cross-polarization has been carried out with a net contact time of ~1 ms at SPINAL-64 decoupling at a MAS spin rate of ~10 KHz. The pulse length for the π/2 pulse at ^1^H has been ~2.5 µs, whereas the average pulse length during the SPINAL-64 decoupling sequence has been ~6.5 µs with 2560 transients, respectively.

#### 2.2.3. Dynamic Light Scattering (DLS) and Zeta Sizer Analyses

The technique allows interpreting the available effective diameter of the as-prepared nano-composites in the wet chemical route. The DLS measurement aids in uncovering the surface charge behavior and diameter of the formation of the nano-composites. The study was further complemented by FESEM analyses. 

In order to examine the surface charge and average diameter of the nano-composites solution (solvent DI water with 2 h sonication), DLS experimentations (Malvern, Westborough, MA, USA) were carried out using flow throughout the system. Technically, Teflon tubing has been connected to a disposable plastic cuvette (which remains stationary to DLS instrumentations). The plastic chamber through which a constant stirred pH adjusted solution whose measurement is to be taken is poured into for measurements. The continuous flow of the nano-composite solution (solvent distilled water) ultrasonicated for 2 h has been stopped during each recording at the rate of 5 min/pattern and ~30 s delay time in order to exchange solution. This step is usually performed for the stoppage in the interference in the process of data acquisition. The overall colloidal morphology has been obtained using the DLS and the zeta–sizer instrumentations.

The methods pertaining the preparation of the DLS studies mentions the standard method of [19,20] In this study, pH adjusted and continuous flow has been mentioned, whereby critically stating the solution of nano-composites (powdered nano-composites dissolved and sonicated into DI water) whose measurements of diameter has been intended for calculations has been pumped in a non-continuous fashion in a regulated time-lapse interval pattern of 5 min/pattern in order to avoid the risk associated with typical SAXS experimentations (whereby continuous flow is required for measurements). Such non-continuous flow of solution in DLS calculations, whereby flow from a fixed plastic cuvette to a plastic beaker is taken using a peristaltic pump mechanism, is performed to avoid the interferences associated with typical data acquisitions required for accurate diameter measurements of the nano-composites in DLS studies. In order to explain the pH adjusted matter, the average neutral pH (to avoid ionic interference in diameter measurements) of the nano-composites has been measured, whereby the powdered nano-composites of AgNP-C60 has been dissolved in the distilled water and ultrasonicated (2 h) before being subjected and measurement to DLS.

#### 2.2.4. Field Emission-Scanning Electron Microscopy (FE-SEM)

This study helps identify the surface topography and morphology of the as-prepared nano-composites in a wet chemical route. The principle of backscattering of electron source from the surface atoms enables the uncovering nature of the surface properties of the as-synthesized nano-composites. The surface morphology of the composites enables one to understand the nature of biophysical interactions at the cell line composite interactions. The surface analyses of the nano-composites provide further confidence to embark into the vertical and inner morphology of the composites. 

In preparing the sample of nano-composites, the powdered sample solutions were filtered at the specific time interval through 0.1 µm polycarbonate filters, which were immediately washed with distilled water for removing the excess salts, impurities and were left for drying at ambient temperature conditions. The filter papers were placed on SEM AI-stubs, coated with 3 nm platinum, and analyzed with a Nova (Nano SEM, 450/FEI, FEI, Lincoln, NE, USA) performed at an acceleration voltage of 15 kV.

#### 2.2.5. High-Resolution Transmission Electron Microscopy (HR-TEM)

The inner morphology of C_60_-AgNPs nano-composites has been easily understood from the HR-TEM analyses, whereby the electron source from the electron gun penetrates deep inside the structure of the composites for the formation of the image at the nanoscale magnification. The study enables to uncover of interesting inner insights about the nano-composites, which has been supported by the Energy Dispersive X-Ray Spectroscopic analyses (EDAX) for uncovering associated elemental compositions of the structure.

The sample of ~4 µL has been drop-casted over the formvar coated-copper grids. It has been dried overnight for examination under HR-TEM at the operational voltage of FEI T20FEG TEM operated at 200 kV.

#### 2.2.6. Antioxidant Activity

The antioxidant activity of C_60_-AgNPs nano-composites was assessed by 1, 1-diphenyl-2-picryl-hydrazil (DPPH) free-radical scavenging [21]. Briefly, the methanol solution of DPPH (1.9 mL) and 0.1 mL of various concentrations of pristine C_60_, pristine AgNPs, and C_60_-AgNPs nano-composites were combined. The reaction solution was then blended vigorously and incubated in the dark conditions for thirty minutes. The absorbance of the reaction solution was taken at 517 nm, and the solution of ascorbic acid with DPPH was considered as a positive control. All experiments were executed in triplicate. The capability of scavenging the DPPH radicals was figured by employing the following equation: DPPH scavenging potential [%] = [AO − A1]/A0] × 100
where AO = absorbance of the controlA1 = sample absorbance


#### 2.2.7. Antimicrobial Activity

To determine the antimicrobial activity of pristine C_60_, pristine AgNPs and the combinations of C_60_-AgNPs nano-composites were tested against both Gram-positive and Gram-negative bacteria strains viz. *Vibrio cholerae*, *Shigella dysenteriae*, *Salmonella typhimurium*, *Escherichia coli*, *Staphylococcus aureus*, *Staphylococcus epidermidis*, *Bacillus licheniformis*, and fungal strain *Candida albicans* adopting agar-well diffusion method [22]. Briefly, the bacterial and fungal inoculums were consistently diffused on a sterile Petri plate consisting of MHA and PDA media, using a sterile disposable spreader. Immediately at post-inoculation, four wells were contrived by a sterile cork borer. Then, the test samples (concentration: 50 μg/mL) were loaded on the pathogen implanted plates by sterile pipettes and incubated for 24–48 h at 37 °C. The standard antibiotics Gentamicin and Clotrimazole were used as a positive control. The zone of inhibition of pristine C_60_, pristine AgNPs, and AgNP-C_60_ nano-composites against the pathogenic strains was determined in mm ± SD. All the experiments were conducted in triplicate under similar conditions, and the zone of inhibitions was measured afterward. 

Moreover, to evaluate more accurate antimicrobial potentials and to evaluate the minimum inhibition concentrations (MICs) of the test samples against the pathogenic strains in terms of IC_50_ values, the micro broth dilution methods were adopted following our previous published experimental procedures [23].

#### 2.2.8. Cytotoxic Activity

The cytotoxicity assay of C_60_-AgNPs nano-composites was conducted using human normal epidermal (HaCaT) and a breast adenocarcinoma cell line, [MDA-MB-231]. The basal medium DMEM enriched with FBS (10%) and streptomycin sulfate andbenzylpenicillin antibiotics at a final concentration of 100 µg/mL and 100 U/mL were used for all cell culture experiments. The cells were incubated at 37 °C and 5% CO_2_ during the experiments. Later, the cells were treated with Trypsin-EDTA (0.25%) at ~75–80% assemblage and implanted in 96-well plates at a density of 5 × 10^3^ cells/well for the 3-(4,5-dimethylthiazol-2-yl)-2,5-diphenyltetrazolium bromide (Methylthiazolyldiphenyl-tetrazolium bromide; MTT) colorimetric assay [2,24].

Cytotoxicity of the cell lines was determined using MTT assay at both 24 and 48 h of incubation with the C_60_-AgNPs nano-composites [25]. The drug doxorubicin was taken as a standard, and MTT (1 mg/mL) in PBS as a stock solution was prepared instantly prior to application. A volume of 500 µL MTT solution of concentration 50 μg/mL was added to each culture plate. The MTT treated cells were incubated for 3 h, and then scaled down and formazan was then separated with 500 μL DMSO. Later, absorbance was calculated at 570 nm in a microplate reader. The viability of the cells was articulated in terms of percentage absorption at treated cells compared to untreated and control cells.

#### 2.2.9. Statistical Analysis

All the assays were carried out in triplicates to ensure the reawakening of the results. The antimicrobial and antioxidant assays results were depicted as a percentage of inhibition, while the cytotoxicity results were depicted as percent viability, comparative to the control. The values gleaned in the assays for the various treatment groups vs. the controls were statistically determined using a Student’s *t*-test (*p* ≤ 0.05).

## 3. Results and Discussion

### 3.1. FTIR Analysis

The intrinsic functional groups and the associated vibrational spectroscopic evidence highlight the observations obtained for the nano-composites of C_60_-AgNPs, as shown in Figure 1. The C-O vibrations of the C60 molecule and the available functional groups prevalent over the AgNPs structure exhibit the nano-composites intermix formation. Further, the vibrational spectroscopic evidence of C-O-H also makes the justification stronger when the intrinsic vibrational groups in the native structure of C_60_ get shifted to form partial shoulder vibrational signatures at C-O indicating the formation of the nano-composites [19]. The vibrational spectroscopic wavenumbers at ~3300 cm^−1^, 1023 cm^−1^, 1400 cm^−1^ are indications for the presence of functional groups for O-H vibrations, C-O and C-H stretching vibrations, respectively [26]. The different stretching vibrations for the available functional groups existing in the basic architecture of fullerene structure and AgNPs structure makes the broad-spectrum dynamics in the observed functional groups understood by the underlying vibrations in the molecules involved in the FT-IR spectrum.

The resulting interactions between the metallic AgNPs and the organic framework of C60 make the vibrational shifts as mentioned hitherto for the C-O, C-H stretching numbers at ~3300 cm^−1^, 1023 cm^−1^, and 1400 cm^−1^. The pristine AgNPs from the previous studies have highlighted the incidence of the vibrations of the functional groups at ~2363 cm^−1^, 1625 cm^−1^, and 1126 cm^−1^, which are the predominant functional stretching vibrational groups in the pristine AgNPs [20]. Pristine C_60_ framework usually shows stretching vibrations for their native carbon structure at ~576 cm^−1^, 1097 cm^−1^, 1181, and 1427 cm^−1^ wavenumbers, respectively [19]. The wavenumber of the pristine C_60_ at 576 cm^−1^ is associated with a primarily radial motion of the carbon atoms, while the other bands at 1181 and 1427 cm^−1^ are associated with a tangential motion of carbon atoms. The most characteristic vibrational mode is the pentagonal ‘pinch’ mode at 1427 cm^−1^ [27,28]. Such spectra of pristine AgNPs and C_60_ are clearly seen to interact among them, resulting in a blend and mixture in their relevant coordination chemistry resulting in the production of shifted and mixed wavenumbers as mentioned hitherto in the section. Such observations correctly justify the formation of the wet procedure mediated synthesis of nano-composites of two different entities of metallic clubbed to the organic framework. The resulting formation of the metal–carbonaceous composites would help explore the intrinsic electronic and electrochemical significance of both the entities in the same platform, which is otherwise elicited independently as studies previously by several authors in biological and electronic applications. 

### 3.2. NMR Spectroscopy

It could be seen from the NMR plot in Figure 2 that the magnetic moment of the as-prepared nano-composites reveals the basic characteristics of AgNPs and the C_60_ molecule. The composite helps in the formation of mixed magnetic alignments in the nanoscaled system. The formation of the nano-composites indicates the formation of interlinked nano-composite architecture, indicating the formation of lattice fringes in the two independent systems. The magnetic moments of the two associated systems are indicated in the form of ppm values, as seen from the figure. It could be understood from the existing NMR spectra that were owing to the induction of surface electrons towards the basic structure of the buckyball fullerenes (C60), the intermixing redox state is producing a native magnetic moment pronounced at ~7.285 ppm against the control peak at~1.577 ppm, respectively [29] in the composite structure, as shown in Figure 2. Such co-mixing phenomenon for the metallic to carbonaceous intermixing is associated with the sharing of outer electrons between the two structures, confirming the formations of such nano-composite structures.

The scanning range of the NMR spectroscopy has been selected at 1–8 ppm, whereby ~7.5 ppm there is an observation of the peak intensity, indicating the formation of magnetic moment characterization for the as-prepared nano-composites, which comprises the predominantly architecture of sp^2^ carbon skeleton of C_60_ molecules. The standard peak at ~1.577 ppm indicates the control peak of ^13^C NMR taken during measurements for comparison. It is understood from the observations that were owing to the predominant peaks at ~1.577 ppm and ~7.7 ppm, the structural entities of the outer electron level from both the interacting elements gets shared in their co-mixed architecture. The structural entanglement of the interacting elements in the nano-composites uncovers the principle of redox state interactions and exchanges, which is assumed to be of π–π interactions between the organic C_60_ molecule and the metallic AgNPs, respectively. It is further assumed that due to the presence and predominance of surface plasmonic electrons [30] over the metallic AgNPs, there is a tendency of the photonic quantization taking place in the interface interactions between the C60 and the metallic AgNPs, resulting in the formation of stable, quantized nano-composites, making it suitable for a myriad of biocidal, bioimaging, and opto-electrochemical applications [31].

### 3.3. Field Emission Scanning Electron Microscopy (FE-SEM)

It could be seen from Figure 3 for the formation of the C_60_-AgNPs nano-composite topographically. The surface topography and surface texture can be easily understood from Figure 3. For example, the depositions of AgNPs in the form of spherical particles over the C_60_ buckyball structure (indicated by agglomerated particles) could be seen in the figure. The image could be corroborated with the respective EDAX analyses, whereby the incidence of a Silver (Ag) peak besides Carbon (C) peak indicates the formation of the nano-composite structures of C_60_-AgNPs.

It could be clearly understood that owing to such elemental depositions of AgNPs over the buckyball architecture (as evident from Figure 3a), that there is a good chance of charge interactions taking place between the AgNPs surface positive groups and the flanking Carbon (C) and oxygen (O) groups in the buckyball fullerene (C_60_) molecule [32]. The surface decoration of the AgNPs over the C_60_ molecule will aid in the overall charge augmentations taking place over the C_60_ molecule, as evident from the nano-composite formations [33]. Such charge repertoire of C_60_-AgNPs nano-composite at the nanoscale would enable different biomedical applications from carrying desired drug payloads to myriad bioelectronic applications.

### 3.4. High-Resolution Transmission Electron Microscopy (HR-TEM)

The internal micrographs of the as-prepared C_60_-AgNPs nano-composites indicated that during the synthesis process, the depositions of the metallic AgNPs over the C_60_ molecule took place at the nanoscale region. The HR-TEM analysis aids in uncovering the nanoscale phenomenon that takes place during the metallic AgNPs deposition over the C_60_ molecules [34,35]. The phenomenon of redox state reactions between the metallic AgNPs and the carbon structured buckyball configuration makes the entire nano-composites a charge repertoire system in parity with our FE-SEM observations, as shown in Figure 3.

As indicated from Figure 4, upon depositions of the metallic AgNPs (seen as agglomerated deposits over the surface of C_60_ molecules). The entire C_60_-AgNPs nano-composites form an electronically mediated system harboring electrostatic charges contributed from the pristine C_60_ molecules and the positive surface defect-induced AgNPs structures. The surface electrons existing over the AgNPs migrate from its surface to the surface of C60 molecules, derived from the charge–charge interactions and surface energy between the metallic-carbon structures, thereby achieving a thermodynamically stable system. Such minimized entropy system of the nano-composites aids in employing the system in different bioelectronic applications.

### 3.5. DLS and Zeta–Sizer Analyses

The size and surface charge are highly accountable for nanomaterials in living cells or any toxicity study in any organisms. Dynamic light scattering spectroscopy gives very accurate results on the average size and apparent charge of any particles of nano ranges. The distribution of average size and apparent charge of the as-prepared C_60_-AgNPs nano-composites in an aqueous solution were calculated using DLS (Malvern, USA). The average size was calculated to be 68.27 ± 0.489 nm (Figure 5a), and the charge was −26.6 ± 4.29 mV (Figure 5b). The particle size plays a vital role in cell transport and communication during its activity against various cellular biosystems. Lesser-sized nanoparticles are flexible for easier movement due to the high surface area of the particles through the membranes of the cells. Hence, the particles ≤100 nm are highly resourceful for applications in drug delivery and the development of biosensing devices [36,37,38]. Similarly, the surface charge of C_60_-AgNPs nano-composites is also an important attribute as it will impact the ability of the nanomaterials to interact with various macromolecules that function in diverse biochemical pathways [39,40].

It could be seen from the figure of DLS and its corresponding zeta analyses that owing to the bottom-up approach of the synthesis of nano-composites of metal-organic framework, the polydispersity, and the granulometry analyses have been strictly analyzed in the present study. The rough edges of the metallic AgNPs and the symmetric C_60_ molecules make the zeta analyses and the DLS a vital parameter for analyses of the as-prepared intercalated nano-composites. It could be seen clearly from the DLS measurements (Figure 5a) that upon shinning a laser light in the instrumentation against the solution of the nano-composites, the different sized nano-composites of calculated ~70 nm were ascertained. It is further ascertained from the DLS measurements of the nano-composites that owing to the bottom-up co-mixation of the participating entities of AgNPs and C_60_ molecules, the resulting DLS of the overall diameter of the nano-composites came to be ~70 nm (68.27 ± 0.489 nm). It is assumed that, albeit the interaction mechanism involved in the formation of the metallic and organic framework of composites in the solvent of distilled water, we ascertain the net average diameter of the composite to be the same as calculated and hence mentioned. The dynamics and kinetics of the as-synthesized nano-composites in the solution are beyond the scope of the present study and would be further analyzed vertically in solution kinetical studies in the future, which would aid to uncover the actual size distribution pattern of the individual nano-composites when compared to its participating individual components. The pristine AgNPs were calculated from the solution phase interaction of the average size distribution to ~15–20 nm and ~50 nm for the C_60_ molecules, which is in close agreement with previous studies [20,41].

Similarly, from the Zeta-sizer analyses, the composite suspension indicates the stability of the as-prepared nano-composites solution. We have achieved a zeta value ~ −26.6 ± 4.29 mV, indicating sparingly stable composite formation. It is understood that the more negative value of the nanosystems indicates the crucial role played by the involved capping agents or associated agents in making the nanoscaled system away from the Reddick threshold [42]. As envisaged previously in the literature, the primary dividing line to establish the particle in terms of more stable or less stable lies in the fact that it falls in the range of +30 to −30 mV value. If the particle suspension falls more than −30 or +30, the resulting suspension is considered more stable. On the other hand, if the resulting suspension of the nanosystem falls in the range of −15 mV, the particle is at the threshold of entering the agglomeration or flocculation. In our case, the surface charge value of the nano-composites is more than −15 mV and within −30 mV. This suspension is at the stage of optimum stability and can be utilized in different biological applications.

### 3.6. Antioxidant Activity

Fullerene (C60) is one of the unique and versatile carbon-based nanomaterials valued in biomedicine development. It acquires remarkable antioxidative capacity, which has fashioned it a strong and major element to be utilized in skin-related cosmetics and anti-aging products [43]. Besides this, AgNPs also have significant contributions in biomedicine and are extensively employed in the development of different antimicrobial cosmetics and other ointments available in the market [44]. The current C60-AgNPs nano-composites materials should have evaluated their safety while triggering use in the human body and other biomedical aspects. Hence, the antioxidant potential was studied in terms of DPPH radical scavenging activity. Current results recorded the scavenging activity of free radicals by C_60_-AgNPs nano-composites was dose-dependent (Figure 6). The scavenging activity was raised as the C60-AgNPs nano-composites’ concentration rose from 25–125 µg/mL. The IC_50_ value was found to be 42.04 ± 0.19 µg/mL. The bequest of hydrogen accomplished the mechanism of DPPH scavenging to the free radicals [45].C_60_ is the most common fullerene, made up of 60 carbon atoms that form a structure that looks like a hollow soccer ball. Because of the way these atoms bond together, C_60_ interacts with free radicals in the environment, giving the molecule strong antioxidant properties [46]. Since oxidative stress plays a crucial role in many therapeutic conditions such as, neurodegenerative disorders, cancers, diabetes mellitus, hypertension, and aging [47,48,49]. Employing fullerene and its composites as an innovative diagnosis and medication have become a promising topic of great significance among biomedical researchers [1,50]. Apart from the fullerene, the AgNPs also have the capacity and performed potential diverse radical scavenging activity. The current findings on DPPH radical scavenging activity are highly relevant to the antioxidant potential shared by the individual entities from fullerene and AgNPs. There are many potential antioxidant reports of different functionalized fullerene and their potential involvement in the biomedicine industry [1,50]. However, the present development of AgNPs functionalized C_60_ nanocomposites is novel and unique to peruse some more innovative applications in the bioindustry. Moreover, such antioxidant potential of C_60_-AgNPs nano-composites will build an excellent opportunity to apply these nanomaterials in diverse fields such as biomedicine and other drug delivery activity.

### 3.7. Antimicrobial Activity

The antimicrobial activity of C_60_-AgNPs nano-composite was preliminarily determined by agar–well diffusion method against seven human pathogenic bacteria *Vibrio cholerae, Shigella dysenteriae, Salmonella typhimurium, Escherichia coli, Staphylococcus aureus, Staphylococcus epidermidis, Bacillus licheniformis,* and fungal strain *Candida albicans* as shown in Table 1. From the zone of inhibition, it was found that C_60_-AgNPs nano-composite has the highest activity against *S. dysenteriae*. Further, the antimicrobial assay was carried out by using the micro broth dilution method to corroborate the results found in the agar–well diffusion method. Table 1 illustrates the effective % inhibition and the respective MICs of each strain. It could be noticed from the agar–well diffusion and micro broth dilution method that Gram-negative *S. dysenteriae* among all the selected bacterial strains exhibited an intensified inhibition % by C_60_-AgNPs nano-composite.

In contrast, five pathogenic bacterial strains exhibited a more than 90% inhibition rate, and the pathogenic fungus also exhibited >90% inhibition. Among the MICs studies, as shown in Table 1, IC_50_ calculations display an evidenced antimicrobial activity of *V. cholera* (105.44 ± 3.67 µg/mL), *S. dysenteriae* (52.72 ± 0.27 µg/mL), *S. typhimurium* (90.10 ± 0.35 µg/mL), *E. coli* (79.54 ± 0.51 µg/mL), *S. aureus* (91.15 ± 0.77 µg/mL), *S. epidermidis* (84.97 ± 0.68 µg/mL), *B. licheniformis*(81.99 ± 1.37 µg/mL), *C. albicans* (76.61 ± 1.98 µg/mL) which has been backed and substantiated with several previous research findings in the similar lines. When Ag NPs and C_60_ were tested separately, Ag NPs showed superior antimicrobial activity than C_60_. Interestingly, the nano-composite C60-AgNPs showed synergistic antimicrobial activities. C_60_-AgNPs nano-composite showed significantly higher antimicrobial activity in comparison to Ag-NPs and C_60_ against all eight tested microorganisms.

From the MICs value, it was also proved that C_60_-AgNPs nano-composites are highly effective against *S. dysenteriae*. Moreover, the as-prepared nanomaterial is also equally potent against other pathogens, which signifies the IC_50_ values. The IC_50_ < 100 µg/mL of the current tested microorganisms endorsed the potential applications of C_60_-AgNPs nano-composite materials as an antimicrobial candidate and as a supporting engagement in developing biomedical devices and other drugs packaging materials. Fungal infections are prevalent and frequently observed in the long-term storage of any stuff, and the anti-candidal activity of such nanomaterials makes it an emerging antifungal nanomaterial that can be used in different bio application fields. There are many reports on the potential antimicrobial activities of both pristine C_60_ and AgNPs. The present findings of enhanced antimicrobial results of these nano-composites make promising hope of profound applications of unique combinatorial nanomaterials organo-metallic in nature. Fullerene has been studied well in potential applications in different cosmetics and other skincare industries due to its hypothesized strong antioxidant activity [43]. Hence now, the adding of promising antimicrobial AgNPs with fullerene can enhance the quality of the products customized using C_60_-AgNPs nano-composite.

The promising antimicrobial actions of C_60_-AgNPs nano-composite may be associated with some well-described mechanisms such as the adhesive capability of C_60_-AgNPs onto the cover of cell wall and membrane, prejudicial activity on major intracellular structures (e.g., mitochondria, ribosomes, vacuoles,) and important biomolecules (e.g., DNA, RNA protein and lipids) due to penetration capability due to nano size, induction of oxidative stress and cellular toxicity due to reactive oxygen species (ROS) generation and harmful free radicals and inflection of signal transduction pathways [51]. Besides these mechanisms, C_60_-AgNPs nano-composite also inflects the human immune system by coordinating inflammatory response [52], which facilitates microorganism’s inhibition.

### 3.8. Cytotoxicity Activity

The HaCaT cells were treated with C_60_-AgNP_s_ nano-composite in the culture at a varied concentration (0.97 nM to 1000 nM) for 24 h at 37 °C. The MTT assay was adopted to determine human keratinocyte (HaCaT) cell viability upon treatment with C_60_-AgNP_s_ nano-composite (Figure 7). The dose-dependent response of C_60_-AgNP_s_ nano-composite on HaCaT cell viability was observed from assay results. HaCaT cells displayed to 125 nM C_60_-AgNP_s_ have shown a 94.26% level of cell viability. The percentage of cell viability, however, slowly reduced as the dose of C_60_-AgNP_s_ raised. Exposure to 1000 nM C_60_-AgNP_s_ dropped off the viability of the HaCaT cells to 75.43%. Current results endorsed that C_60_-AgNP_s_ are comparatively non-toxic to normal cells as >50% of the cells were viable, even at a higher dose. Hence, it materializes that C_60_-AgNP_s_ nano-composite can be used for important diverse applications due to its biocompatibility property. The biocompatibility of C_60_-AgNP_s_ with HaCaT cells has also been manifested in earlier studies [53,54].

The cell viability of C_60_-AgNPs nano-composite treated breast cancer cells (MDA-MB-231) was accessed in a similar manner. In contrast to results with normal cells (HaCaT), the cytotoxicity assay of C_60_-AgNPs nano-composite with MDA-MB-231 revealed axiomatically higher levels of cytotoxicity (Figure 8 and Appendix A). The interaction of the cancer cells with a ~1000 nM C_60_-AgNPs nano-composite for 24 h lowered the cell viability to ~20.46 %. A time and dose-dependent cytotoxicity of C_60_-AgNPs nano-composite against MDA-MB-231 cells has been reported earlier [55,56,57].

The percentage of viable MDA-MB-231 cells was reduced as the dose of C_60_-AgNPs was raised. The percentage of viable cells was 62.86% and 99.29% after interaction with 250 nM and 0.97 nM C_60_-AgNPs solutions, reciprocally. The present results proved that MDA-MB-231cells displayed a dose-dependent response in cytotoxic activity study. However, the cytotoxicity did not occur to be time-dependent as a very minute difference in the percentage of viability was observed after both 24 and 48 h of incubation, maintaining the similar concentration of C_60_-AgNPs (Figure 8). The current observation results suggested that reactive oxygen species (ROS) generation plays a significant role in the cellular toxicity mechanism of C_60_-AgNPs in breast adenocarcinoma cells [56]. The crucial organelles, e.g., mitochondrion, also deceive the toxicity of C60-AgNPs inside cellular systems, and the ROS induces cellular apoptosis when C_60_-AgNPs nanomaterials are brought into a cell and disarray the whole metabolism [55]. During the treatment with doxorubicin (control), the IC_50_ values are lower in MDA-MB-231 cells than HaCaT cells after 24 h and 48 h of susceptivity. Moreover, the IC_50_ values for the C_60_-AgNPs treatment were lower in MDA-MB-231 cells than in HaCaT cells after both 24 h and 48 h of susceptivity (Figure 8). The IC_50_ values of C_60_-AgNPs compared to doxorubicin strongly proved to be the usage in cancer therapy in biomedical sectors.

## 4. Conclusions

The study helps to explore the various physico-chemical features and bioactivities of the metallic AgNPs and C_60_ nano-composites. Diverse spectroscopic and micro graphical techniques have been used to uncover the electronical and chemical behavior of the synthesized nano-composites, possessing organic carbon-based entities as well as their metallic counterparts in its native structure. Moreover, when the nano-composite was assorted with cell lines of MDA-MB-231, it elicited a meagre cytotoxicity behavior. Such a unique combination of zero-dimensional carbon framework systems possessing buckyball structure, ingrained with a spherical nano-particulate system having huge surface plasmonic resonance, surface energy, and surface area attributes. Thus, this would be a novel approach to mitigating several bioelectronics issues to biomedical applications.

## Figures and Tables

**Figure 1 ijms-23-00714-f001:**
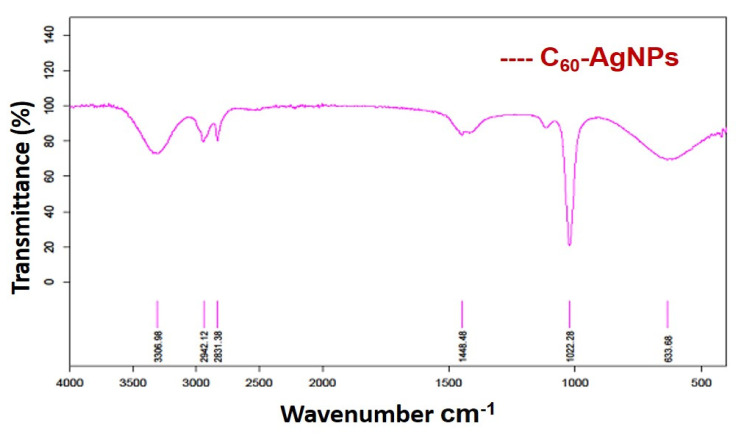
FT-IR of C_60_-AgNP nano-composite indicating different stretching vibrations and associated functional groups.

**Figure 2 ijms-23-00714-f002:**
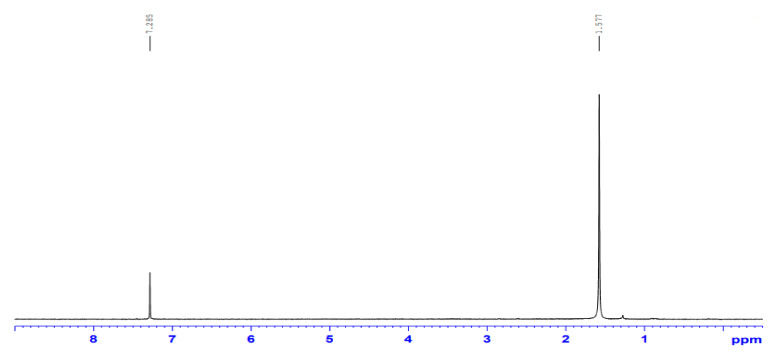
^13^C NMR of C_60_-AgNP nano-composite.

**Figure 3 ijms-23-00714-f003:**
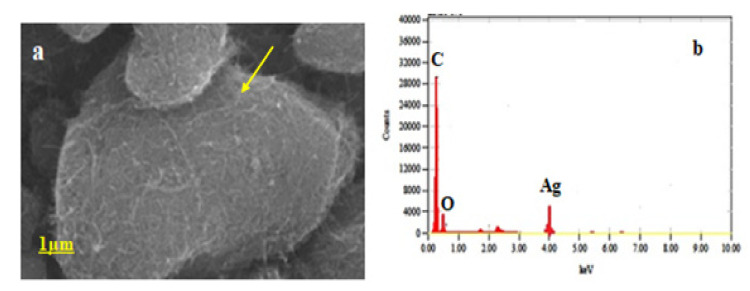
(**a**) FE-SEM of C_60_-AgNPs nano-composite viewed from the topical angle (shown by arrow); (**b**) EDAX of C_60_-AgNPs composite indicating the presence of elemental Silver (Ag), Carbon (C), and Oxygen (O) groups, respectively.

**Figure 4 ijms-23-00714-f004:**
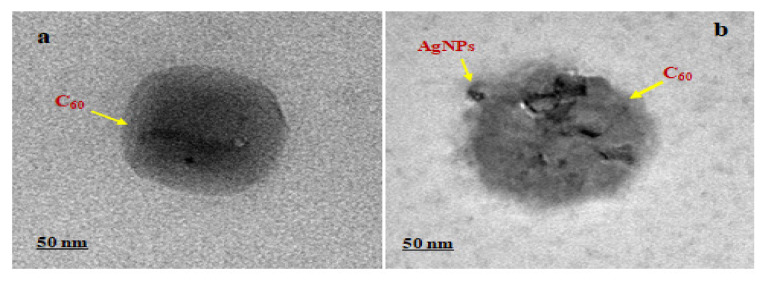
(**a**) HR-TEM micrograph of C_60_ molecule; (**b**) C_60_-AgNPnanocomposites, the deposits of AgNPs over the surface of C_60_ molecule could be seen; indicated by arrows.

**Figure 5 ijms-23-00714-f005:**
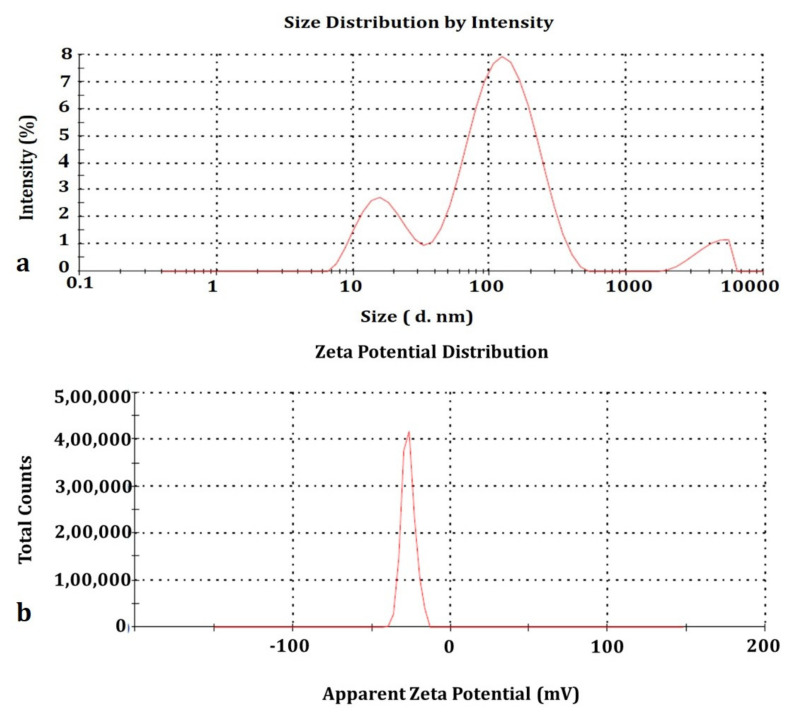
DLS analysis of C_60_-AgNPs nano-composites (**a**) Average Size distribution (**b**) Surface charge.

**Figure 6 ijms-23-00714-f006:**
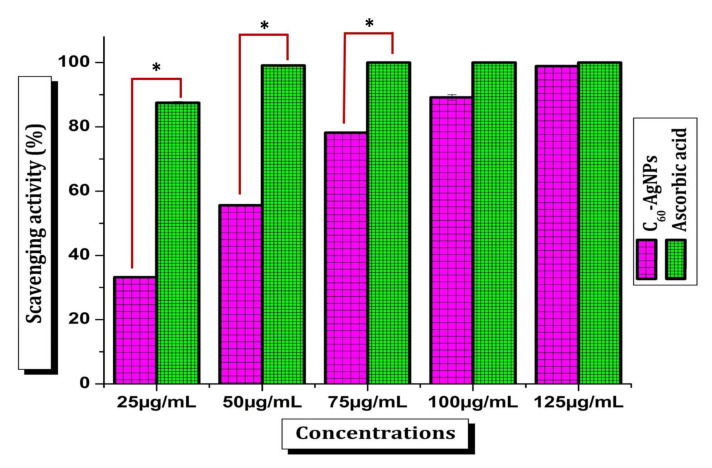
DPPH radical scavenging assay by C_60_-AgNPs nano-composites. Error bar represents the standard deviation of the mean. * *p* ≤ 0.05. Significant difference (*p* ≤ 0.05) within a parameter between two lines is denoted by an asterisk.

**Figure 7 ijms-23-00714-f007:**
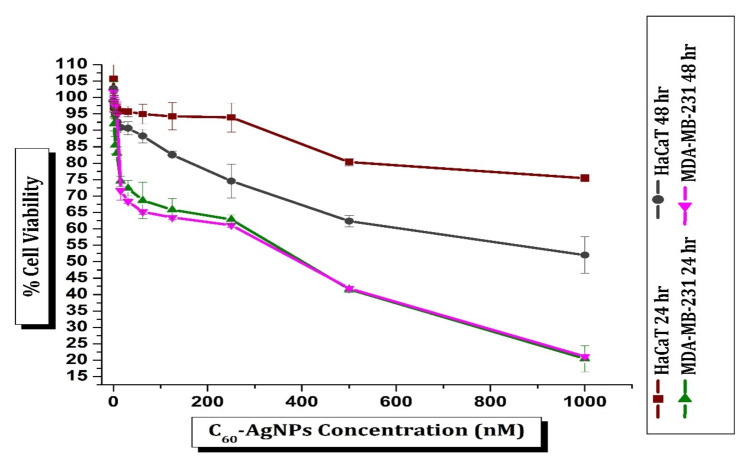
Cell viability of MDA-MB-231 and HaCaT after treatment with different concentrations of C_60_-AgNP_s_ nano-composite after 24 h and 48 h.

**Figure 8 ijms-23-00714-f008:**
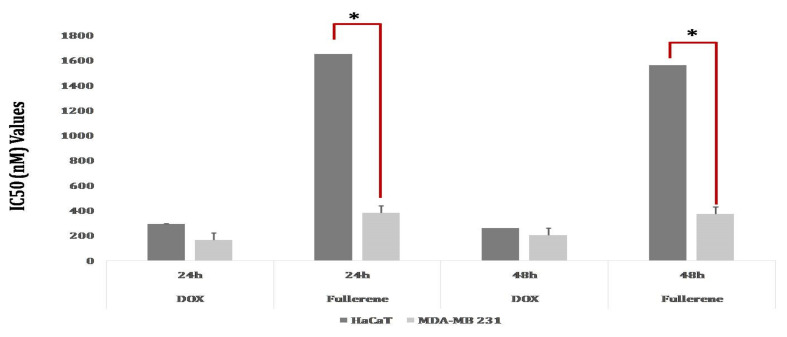
Cell viability [IC_50_ value] of MDA-MB-231andHaCaT, after treatment with different nano-formulations [Dox: Doxorubicin [control], C_60_-AgNPs nanocomposite] after 24 h and 48 h. Error bar represents the standard deviation of the mean. * *p* ≤ 0.05. Significant difference (*p* ≤ 0.05) within a parameter between two lines is denoted by an asterisk.

**Table 1 ijms-23-00714-t001:** Antimicrobial activity of C_60_-AgNPs nano-composite.

Name of the Test Strain	Mean Zone of Inhibition ± SD (in mm)			Percentage of Inhibition (%) ± SD	MIC IC_50_ (µg/mL)
	C_60_	AgNPs	C_60_-AgNPs	C_60_-AgNPs	C_60_-AgNPs
*Vibrio cholerae*	12.2 ± 0.08	11.4 ± 0.04	13.10 ± 0.04	84.97 ± 0.68	105.44 ± 3.67
*Shigelladysenteriae*	13.3 ± 0.00	16.3 ± 0.00	21.20 ± 0.00	99.30 ± 0.37	52.72 ± 0.27
*Salmonella typhimurium*	12.6 ± 0.00	13.1 ± 0.00	17.60 ± 0.00	92.43 ± 0.09	90.10 ± 0.35
*Escherichia coli*	13.33 ± 0.05	15.23 ± 0.05	18.67 ± 0.05	93.37 ± 0.76	79.54 ± 0.51
*Staphylococcus aureus*	11.57 ± 0.05	13.47 ± 0.05	14.87 ± 0.05	86.60 ± 0.73	91.15 ± 0.77
*Staphylococcus epidermidis*	12.57 ± 0.00	13.10 ± 0.00	15.23 ± 0.05	92.03 ± 0.48	84.97 ± 0.68
*Bacillus licheniformis*	13.37 ± 0.05	14.40 ± 0.00	16.20 ± 0.00	92.80 ± 0.57	81.99 ± 1.37
*Candida albicans*	12.60 ± 0.00	13.80 ± 0.00	17.50 ± 0.00	94.90 ± 0.54	76.61 ± 1.98

## Data Availability

The data supporting the reported results are available upon request from the corresponding author.

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
