# Peer review of "Exploring the Bioactive Potentials of C60-AgNPs Nano-Composites against Malignancies and Microbial Infections"

_ijms, 2022, doi:10.3390/ijms23020714_

Round 1

Reviewer 1 Report

In this manuscript, the antimicrobial and antioxidant activities of C60-AgNPs nanocomposites have been evaluated over the bio-systems in-vitro including human breast adenocarcinoma (MDA-MB-231) cells and normal human epidermal (HaCaT) cells. The experimental data indicates that the C60-AgNPs nanocomposites elicit clearly cytotoxicity towards MDA-MB-231 cell. However, there are several issues which should be addressed before publication. Specific comments,

  • As shown in Figure 4 (a), the size of C60 is larger than 100 nm. The authors should check the TEM micrograph carefully, and explain the phenomenon.
  • The average sizes of AgNPs and C60-AgNPs nanocomposites should be calculated by TEM measurement. The average size (68.27±0.489 nm) of C60-AgNPs nanocomposites by DLS measurement is smaller than the size of C60 in Figure 4 (a). It is strange.
  • There is error in Figure 8. Is there any difference between Figure 7 and Figure 8?
  • Is the antimicrobial activity of C60-AgNPs nanocomposite better than that of AgNPs?
  • The mass ratio of AgNP to C60 in the C60-AgNPs nanocomposite should be evaluated.

Author Response

Plz find the attached file 

Reviewer 2 Report

The experimental data were insufficient to support the conclusion that the study helps to uncover the different physico-chemical properties of the metallic AgNPs and C60 nanocomposites.

Also, the spectroscopic and micro graphical analyses can barely aid in uncovering the underlying electronical and chemical behavior of the as-synthesized nanocomposites bearing organic carbon-based entities and their metallic counterparts in its native structure.

The results this study obtained were not adequate to cover the title “physico-chemical, biophysical of C60-AgNPs nanocomposites over biosystems”. And the principles behind the antimicrobial and anticancer effect of the C60-AgNPs nanocomposites were not well demonstrated.

Author Response

Q1) The experimental data were insufficient to support the conclusion that the study helps to uncover the different physico-chemical properties of the metallic AgNPs and C60 nanocomposites.

Ans: Thanks to the reviewer for such critical evaluator comments; In fact the authors felt the characterization is enough to justify the formation of nanocomposites between the metallic nanoparticles over the carbon molecules. Moreover, at the best of their discretion and capacity, authors tried to analyse and verify different anti-oxidant, anti microbial and cytotoxic analyses for the as prepared nanocomposies in the in-vitro conditions.

Moreover, the authors will try to improve the articles if reviewer will point out the specific need of any more study on the nano-composites.

Q2) Also, the spectroscopic and micro graphical analyses can barely aid in uncovering the underlying electronically and chemical behaviour of the as-synthesized nanocomposites bearing organic carbon-based entities and their metallic counterparts in its native structure.

Ans: Thanks for the reviewers suggestion and valuable observations. In the situation of pandemic, the authors have tried their level best to aid the characterisations of the formation of the nano-composites using spectroscopic techniques like FT-IR, NMR spectroscopy etc. Moreover, HR-TEM and FE-SEM analyses along with their respective EDAX analyses helps in understanding the formation of the nano-composites both qualitatively and quantitatively.

Q3) The results this study obtained were not adequate to cover the title “physico-chemical, biophysical of C60-AgNPs nanocomposites over biosystems”. And the principles behind the antimicrobial and anticancer effect of the C60-AgNPs nanocomposites were not well.

Ans: Thanks for the valuable suggestion of the reviewers. The title “physico-chemical, biophysical of C60-AgNPs nanocomposites over biosystems” as per the best understanding of the author is concerned that physico-chemical characterisation is well established in the materials and method portion of the characterisation of the ‘prsitine’ and ‘nano-composites’ is concerned. The spectroscopic and optical support as mentioned in Section 3.1 to 3.5 for FT-IR to NMR characterisations explains the validation of the physico-chemical characterisations of the as prepared nano-composites. Next is the biophysical portion has been well explained and mentioned in the Section 3, for different biological nano-composites formation like FTIR, HR-TEM (with EDAX) and associated biological analyses.

 The antimicrobial activity of C60-AgNPs nanocomposite was evaluated qualitatively (agar well diffusion method for screening purpose) and then quantitatively (broth dilution method). The authors performed extensive antimicrobial activity study against wide range of bacteria (both gram positive and gram negative) and fungi. The estimation of IC50value is most accurate measure of antimicrobial activity. Moreover, the as-prepared nanomaterial (C60-AgNPs nanocomposites) was also equally potent against other pathogens, which signifies the IC50 values.Interestingly, the nanocomposite C60-AgNPs showed synergistic antimicrobial activities. C60-AgNPs nanocomposite showed significant higher antimicrobial activity in comparison to Ag-NPs and C60 against all 8 tested microorganisms.

Again, anticancer activity of C60-AgNPs nanocomposites was studied against both normal (HaCaT cell) and cancer cells (MDA-MB-231 cells). From quantitative analysis, the dose-dependent response of C60-AgNPs nanocomposite on HaCaT cell viability was observed.In contrast to results with normal cells (HaCaT), the cytotoxicity assay of C60-AgNPs nanocomposite with MDA-MB-231 revealed axiomatically higher levels of cytotoxicity. During the treatment with doxorubicin (control), the IC50 values are lower in MDA-MB-231 cells than HaCaT cells after 24 hours and 48 hours of susceptivity. Moreover, the IC50 values for the C60-AgNPs treatment were lower in MDA-MB-231 cells than in HaCaT cells after both 24 hours and 48 hours of susceptivity. The IC50 values of C60-AgNPs compared to doxorubicin strongly proved to be the usage in cancer therapy in biomedical sectors.

With respect to the reviewer's multiple progressive suggestions, the title of the manuscript has been revised and highlighted in yellow colour.

Reviewer 3 Report

The manuscript entitled "Uncovering the physico-chemical, biophysical and cytotoxic analysis of C60-AgNPs nanocomposites over biosystems" is well written with a reasonable set of experimental designs. Indeed, the manuscript required revision considering the following suggestions. 

  1. It is suggested to include the FT-IR spectra of C60 and AgNP also and improve the discussion of C60-AgNP nanocomposite in comparison to spectra of individual components (such as C60, and AgNP). It will be interesting for the reader to know their comparative analysis.
  2. It is suggested to include the 13C NMR spectra of C60 and AgNP also and improve the discussion of C60-AgNP nanocomposite in comparison to spectra of individual components (such as C60, and AgNP). It will be interesting for the reader to know their comparative analysis.
  3. It is suggested to include the FE-SEM and EDAX spectra of C60 and AgNP also and improve the discussion of C60-AgNP nanocomposite in comparison to spectra of individual components (such as C60, and AgNP). It will be interesting for the reader to know their comparative analysis.
  4. It is suggested to include the HR-TEM spectra of AgNP also and improve the discussion of C60-AgNP nanocomposite in comparison to spectra of individual components (such as C60, and AgNP). It will be interesting for the reader to know their comparative analysis.
  5. It is suggested to highlight the strategy utilized to make the aqueous sample of C60-AgNP nanocomposite remain in a dispersed state. In my opinion, the sample will be settled down in the bottom instantly in addition to water for DLS and zeta potential analysis. 
  6. It is suggested to include the PDI value of C60-AgNP nanocomposite during DLS analysis. Figure 5a indicates a polydisperse system of  C60-AgNP nanocomposite. Generally, a monodisperse system is preferable and has better stability. Considering this fact improve your discussion and elaborate your effort try to make it a monodispersive system of C60-AgNP nanocomposite.
  7. It is suggested to include the particle size and zeta potential spectra of C60 and AgNP also and improve the discussion of C60-AgNP nanocomposite in comparison to spectra of individual components (such as C60, and AgNP). It will be interesting for the reader to know their comparative analysis.
  8. The author has performed a comparative analysis of the antimicrobial activity of C60, AgNP, and C60-AgNP nanocomposite. This comparative analysis is interesting to signify the rationale of C60-AgNP nanocomposite preparation. Therefore, I highly recommend including the antioxidant and cytotoxicity analysis for C60, and AgNP also. It will be interesting for the reader to know their comparative analysis findings. 

Author Response

Comments and Suggestions for Authors

The manuscript entitled "Uncovering the physico-chemical, biophysical and cytotoxic analysis of C60-AgNPs nanocomposites over biosystems" is well written with a reasonable set of experimental designs. Indeed, the manuscript required revision considering the following suggestions. 

  1. It is suggested to include the FT-IR spectra of C60 and AgNP also and improve the discussion of C60-AgNP nanocomposite in comparison to spectra of individual components (such as C60, and AgNP). It will be interesting for the reader to know their comparative analysis.

Ans: Authors thanks the reviewer for the valuable suggestion. In the revised manuscript, authors have tried to provide the relevant spectroscopic evidences in terms of the comparative functional groups existing in the native structure of pristine C60, pristine AgNPs and the significant alterations in the corresponding functional groups upon co-mixing phenomenon taken place during bottom-up nano-composite preparation of C60-AgNP nanocomposites in the ambient laboratory conditions. Individual plots of FT-IR for each participating nano-components are not available and are therefore presented in the combined form in Fig. 1 and has been explained and discussed elaborately in Sec 3.1, the existing vibrational molecular interactions among the individual nano-components in the composites, explaining thereby the successful formation of the as-desired metallic-organic nanocomposites.

  1. It is suggested to include the 13C NMR spectra of C60 and AgNP also and improve the discussion of C60-AgNP nanocomposite in comparison to spectra of individual components (such as C60, and AgNP). It will be interesting for the reader to know their comparative analysis.

Ans: Authors thanks the reviewer for the valuable suggestion. In order to uncover the existing magnetic moment interactions playing between the metallic AgNPs and the organic C60 molecules, authors have tried to explain the combined nature of the magnetic moments and the structural interactions taking place between the participating molecules and is discussed in detail in Section 3.2. The relevant interactions taking place in terms of the resonating interactions between the two molecules is well shown in Fig. 2. As per suggestion of reviewer, individual plots of each components of AgNPs and C60 are not given in the manuscript as only combined nano-composites 13C NMR spectra has been performed. It is highly anticipated that the reviewer would be convinced with the existing explanations of the said measurement as provided in Section 3.2.

  1. It is suggested to include the FE-SEM and EDAX spectra of C60 and AgNPs also and improve the discussion of C60-AgNP nanocomposite in comparison to spectra of individual components (such as C60, and AgNPs). It will be interesting for the reader to know their comparative analysis.

Ans: Authors thanks the reviewer for the valuable suggestion. As per suggestion of the reviewer, authors have provided the individual FESEM micrographs of C60 and AgNPs respectively in Fig 3 (a) and Fig 3 (b). The individual FESEM micrographs of metallic AgNPs and organic C60 molecule indicates the successful formation of the participating nano-components of the nano-composites. Individual pristine AgNPs formation and C60 formation is well seen from the figures. It explains the verification that using bottom-up chemical mediated route, the monodisperse particles of average diameter~ 15-20 nm of AgNPs were formed and average diameter ~ 45-50 nm of C60 molecules has been synthesized in the laboratory situations. EDAX elemental justification of the formation of the nano-composites has been clearly shown in Fig 3(b). Thus, authors have not individually calculated the EDAX analyses of the individual participating nano-components of the nano-composites.

  1. It is suggested to include the HR-TEM spectra of AgNP also and improve the discussion of C60-AgNP nanocomposite in comparison to spectra of individual components (such as C60, and AgNP). It will be interesting for the reader to know their comparative analysis.

Ans: As per suggestion of the reviewer, authors have provided the individual HR-TEM of pristine AgNPs for betterment of the understanding of the existing HR-TEM provided in the Fig 4(b). The lattice d-spacings of the metallic AgNPs came to be ~ 0.137 nm, which has been calculated using appropriate grain size measurement and is came to tbe polycrystalline in nature.

  1. It is suggested to highlight the strategy utilized to make the aqueous sample of C60-AgNP nanocomposite remain in a dispersed state. In my opinion, the sample will be settled down in the bottom instantly in addition to water for DLS and zeta potential analysis. 

Ans: Authors thanks the reviewer for the suggestion. In the best understanding of the authors, a conventional simple co-mixing bottom up solution route as mentioned in Section 2.2 of materials section has been mentioned following reference [18] of the preparation steps. In order to better understand and analyze the zeta measurements and DLS calculations, the prepared nano-composites powders were solubilized in distill water (deionized water) and has been sonicated for 15-20 minutes, followed by isolating the supernatant of the solution mixture. As rightly suggested and pointed by the reviewer, upon addition of water to the powder composites, after few minutes of supernatant and vertexing, the resulting mixture tries to settle at the bottom. We have carefully isolated the upper supernatant solvent upon mixing, for diluted sample collections which is highly recommended for sensitive diameter and surface charge measurements of lower scaled materials like in our case nano-composites.

  1. It is suggested to include the PDI value of C60-AgNP nanocomposite during DLS analysis. Figure 5a indicates a polydisperse system of C60-AgNP nanocomposite. Generally, a monodisperse system is preferable and has better stability. Considering this fact improve your discussion and elaborate your effort try to make it a monodisperse system of C60-AgNP nanocomposite.

Ans: Authors thanks the reviewer for the suggestion. In the best understanding of the surface charge and diameter measurements of the as-synthesized nano-composites. The average diameter came to be ~ 68.27±0.489 nm and the average charge was -26.6±4.29 mV. It could be well understood from the DLS measurements that, upon bottom-up solution-based synthesis of nano-composites, the reaction mixture, solvents, reductants etc plays a critical role in determining the overall stability of the system as a whole. It is assumed that the resulting nano-composites exhibits an average polydispersity nature of the system. Further analyses are required to stabilise the system in terms of using capping agents like CTA, PLGA etc for producing free standing stable nano-composites, which is currently out of the scope of the present studies. Poly Dispersity Index (PDI) has not been studied in the present analyses after synthesis and is currently not feasible.

  1. It is suggested to include the particle size and zeta potential spectra of C60 and AgNPs also and improve the discussion of C60-AgNP nanocomposite in comparison to spectra of individual components (such as C60, and AgNPs). It will be interesting for the reader to know their comparative analysis.

Ans: As per the suggestion of the reviewer, the nano-composites of C60-AgNPs have been shown in Fig. 5. The DLS and the Zeta surface charge analyses clearly explains the nano-composites average charge and diameter of the as-prepared nano-composites. During synthesis of the nano-composites solution, authors have perceived that the charge transfer mechanism in the nano-composites average diameter and the surface charge would be sufficient to explain the role of participating nano-entities in determining and governing the resulting zeta charge and diameter of the composites formed and hence individual DLS and Zeta measurements of the participating nano-components have not been provide by the authors in the manuscript.

  1. The author has performed a comparative analysis of the antimicrobial activity of C60, AgNP, and C60-AgNP nanocomposite. This comparative analysis is interesting to signify the rationale of C60-AgNP nanocomposite preparation. Therefore, I highly recommend including the antioxidant and cytotoxicity analysis for C60, and AgNP also. It will be interesting for the reader to know their comparative analysis findings. 

Ans: In the best understanding of the authors, the cytotoxicity, antimicrobial and anti-oxidant activities of the as-prepared nano-composites have been shown by the authors. In the authors view the biological activities elicited by the nano-composites of C60-AgNPs is sufficient to explain the individual role of the participating nano-entities (C60 and AgNPs) towards the biological targets. The present paper highlights predominantly the underlying mechanisms playing for the as-prepared nano-composites against a suitable target, which has been shown meticulously by the authors using different biological assays. Individual cytotoxicity and associated biological assays of individual C60 and AgNPs are beyond the scope of the present study and hence mechanism of assays of as-prepared nano-composites against the selected biosystems has been embarked in this study.

Authors are highly thankful to the Reviewer 3 for his consistent willing ness in reading our article thoroughly to give constructive comments in improving the articles; but Authors have tried their best to compliance and improve the articles as per the scope and aim of the special theme and necessary applications.

Round 2

Reviewer 1 Report

The revised manuscript could be published as it.

Author Response

Thanks Reviewer for your more attentive comments to improve the manuscripts and the English language of the manuscript has been corrected and improved which can be easily dectable in track change mode.

The modified article is attached in the attached section.

Reviewer 3 Report

The current version of the manuscript should be accepted for publication in IJMS.

Author Response

English language and style are fine/minor spell check required

Thank you Respected Reviewer for your more thorough comments on this manuscript to make it more improved one. As per the comments, the MS has been revised more with reference to some language corrections.

"Please see the attachment
